# A Zero-Dimensional Organic Lead Bromide of (TPA)_2_PbBr_4_ Single Crystal with Bright Blue Emission

**DOI:** 10.3390/nano12132222

**Published:** 2022-06-28

**Authors:** Ye Tian, Qilin Wei, Hui Peng, Zongmian Yu, Shangfei Yao, Bao Ke, Qiuyan Li, Bingsuo Zou

**Affiliations:** 1Beijing Key Laboratory of Nanophotonics & Ultrafine Optoelectronic Systems, Beijing Institute of Technology, Beijing 100081, China; tianye080t@163.com; 2Guangxi Key Lab of Processing for Nonferrous Metals and Featured Materials and Key Lab of New Processing Technology for Nonferrous Metals and Materials, Ministry of Education, School of Resources, Environments and Materials, Guangxi University, Nanning 530004, China; qlwei@st.gxu.edu.cn (Q.W.); yzmmaterials@163.com (Z.Y.); yaoshangfei@st.gxu.edu.cn (S.Y.); abnerkeb@163.com (B.K.); liqiuyan63@163.com (Q.L.)

**Keywords:** self-trapped exciton, 0D organic metal halides, photoluminescence mechanism, blue emission, excited state structure distortion

## Abstract

Blue-luminescence materials are needed in urgency. Recently, zero-dimensional (0D) organic metal halides have attractive much attention due to unique structure and excellent optical properties. However, realizing blue emission with near-UV-visible light excitation in 0D organic metal halides is still a great challenge due to their generally large Stokes shifts. Here, we reported a new (0D) organic metal halides (TPA)_2_PbBr_4_ single crystal (TPA^+^ = tetrapropylammonium cation), in which the isolated [PbBr_4_]^2−^ tetrahedral clusters are surrounded by organic ligand of TPA^+^, forming a 0D framework. Upon photoexcitation, (TPA)_2_PbBr_4_ exhibits a blue emission peaking at 437 nm with a full width at half-maximum (FWHM) of 50 nm and a relatively small Stokes shift of 53 nm. Combined with density functional theory (DFT) calculations and spectral analysis, it is found that the observed blue emission in (TPA)_2_PbBr_4_ comes from the combination of free excitons (FEs) and self-trapped exciton (STE), and a small Stokes shift of this compound are caused by the small structure distortion of [PbBr_4_]^2−^ cluster in the excited state confined by TPA molecules, in which the multi-phonon effect take action. Our results not only clarify the important role of excited state structure distortion in regulating the STEs formation and emission, but also focus on 0D metal halides with bright blue emission under the near-UV-visible light excitation.

## 1. Introduction

Due to their unique photophysical properties, 0D organic metal halides play an important role in many fields, such as light-emitting diodes (LEDs), solar cells, photodetectors [1,2,3,4,5,6]. In 0D organic metal halides, anionic metal halide polyhedron is encapsulated and completely isolated by organic cations, thus forming a unique “host-guest” structure. As a result of the spatial constraints of isolated metal halides, the emission of 0D organic metal halides is caused by the radiation relaxation of local excitons, and generally exhibits efficient broadband emission with a large Stokes shift upon photoexcitation [7,8,9]. These remarkable characteristics make it an ideal candidate material for solid-state lighting. However, most of the energy dissipated in the form of vibration due to the large Stokes shift, making it a great challenge to achieve blue emission in 0D metal halides [3,10,11,12].

Due to their structural diversity and excellent optical properties, lead-based organic metals become one of the most studied 0D hybrid materials. For example, (C_9_NH_20_)_6_Pb_3_Br_12_ exhibits a green emission and a Stokes shift of 151 nm [5]. (N-MEDA)[PbBr_4_] shines a white emission band with a Stokes shift of 170 nm [13]. Obviously, the large excited state structure distortion of the above 0D organic metal halides under photoexcitation leads to their exhibition of a large Stokes shift and a wide FWHM [14]. Therefore, the realization of blue emission by reducing stoke shift is of great significance in the study of 0D hybrid materials.

Of course, there are also some reports of 0D metal halides with blue emission [15]. For example, Zhou et al. synthesized a 0D metal halide of (C_9_NH_20_)_7_(PbCl_4_)Pb_3_Cl_11_ with blue emission band at 470 nm excited by 350 nm UV light sources, and a Stokes shift of 120 nm [16]. Sun et al. reported a 0D metal halide of [BAPrEDA]PbCl_6_·(H_2_O)_2_ with broadband blue emission peaking at 392 nm excited by 300 nm UV light sources and a Stokes shift of 90 nm [17]. However, in order to obtain 0D metal halides with blue emission, the excitation wavelength needs to have higher energy and the excitation peak is generally less than 350 nm, which is affected by the large Stokes shift of 0D metal halide. Therefore, considering most of these 0D organic metal halides with blue emission need to be excited by harmful high-energy UV light sources, and there is a huge energy loss between excitation and emission, these compounds are unsuitable for use in solid-state lighting.

Here, we report a new compound of lead-based (TPA)_2_PbBr_4_ single crystal (SCs), which shows a blue emission band at 437 nm with a relatively small Stokes shift of 53 nm and a narrow FWHM of 50 nm. The photophysical mechanism of (TPA)_2_PbBr_4_ SCs was discussed via temperature-dependent PL spectra, temperature-dependent Raman spectra and DFT calculation, which indicated that the observed blue emission in (TPA)_2_PbBr_4_ comes from the combination of FEs and STE. The theoretical calculation shows that the small excited state structure distortion is the dominant reason for the relatively small Stokes shift in this compound. Therefore, our results not only deepen the understanding of the relationship between structure distortion and the emission characteristic, but also provide some new ideas for the design of high-performance metal halides with blue emission.

## 2. Experiment Section

### 2.1. Materials 

Lead bromide (PbBr_2_, 99.95%), tetrapropylammonium bromide (TPABr, 99%), and N, N-Dimethylformamide (99.5%) were purchased from Shanghai Aladdin Bio-Chem Technology Co., LTD (Shanghai, China) and used as received. 

### 2.2. Synthesis of (TPA)_2_PbBr_4_ Single Crystals (SCs)

The 2.5 mmol PbBr_2_ and 5 mmol TPABr were fully dissolved in N, N-Dimethylformamide at room temperature (RT). Subsequently, the solution was volatilized slowly in the hot stage of 60 °C. After 24 h, the bulk SCs of (TPA)_2_PbBr_4_ can be harvested. 

### 2.3. Characterization

The crystal structure information of (TPA)_2_PbBr_4_ was performed on Bruker D8 Quest X-ray single crystal diffractometer at 298 K. The powder X-ray Diffraction (PXRD) data were collected by Bruker Advance D8 diffractometer with Cu-Kα radiation. The photoluminescence (PL) and PL excitation (PLE) spectra were collected by HORIBA FluoroMax+ instrument. The absorption spectrum was conducted by Shimadzu UV-3600 spectrophotometer. Decay lifetimes and photoluminescence quantum efficiency (PLQE) were obtained by Edinburgh Instruments of FLS980. Variable-temperature PL spectra were also collected by Horiba instrument with the excitation wavelength at 365 nm. The Raman spectra were collected by LabRAM HR Evolution. Thermal stability was measured by TA discovery instrument in nitrogen atmosphere.

### 2.4. Calculation Details

The electronic structure of (TPA)_2_PbBr_4_ was calculated via Density Functional Theory (DFT) calculations using the Vienna Ab initio simulation package (VASP) [18]. The generalized gradient approximation of the Perdew–Burke–Ernzerhof (PBE) [19,20] parameterization with projector-augmented wave [21] method is performed for the exchange and correlation functional. Experimental structure information of (TPA)_2_PbBr_4_ was directly used for the theoretical calculations. Ultra-soft pseudopotentials are utilized for all elements, including C, H, N, Pb, and Br. 400 eV, 1.0 × 10^−5^ eV and 0.01 eV/Å were chosen as the cutoff energies for the plane-wave basis, the self-consistent total-energy difference, and the convergence criteria for forces on atoms, respectively. In the first Brillouin zone, a Monkhorst-Pack k-mesh grid of 4 × 4 × 2 is used.

## 3. Results and Discussion

Bulk (TPA)_2_PbBr_4_ SCs were synthesized by simple solution synthesis method. Then, we characterized its crystal structure information by SCXRD and the detailed crystal structure parameters are given in Appendix A. Clearly, (TPA)_2_PbBr_4_ was composed of isolated [PbBr_4_]^2−^ clusters ionically bonded with TPA^+^ (Figure 1a), resulting a typical 0D structure. (TPA)_2_PbBr_4_ shows monoclinic *I2/a* symmetry with the cell parameters of *a* = 14.93 Å, *b* = 14.51 Å, *c* = 31.93 Å, and *V* = 6895.2 Å. Moreover, each Pb atom is coordinated with four Br atoms, thus forming a unique tetrahedral structure (Figure 1b), which is relatively rare. The bond distance of Pb–Br ranges from 2.7241 to 3.0042 Å, while the angle of Br–Pb–Br varies from 97.04° to 126.18° (Figure 1c). It is worth noting that (TPA)_2_PbBr_4_ has excellent environmental stability. As shown in the Appendix A, the PXRD pattern of the sample stored in air for one month has a similar profile to that of the pristine one. In addition, it is stable under high-intensity UV irradiation (Appendix A). We found that its intensity could still maintain 95% of the initial intensity after high-intensity ultraviolet irradiation for 300 min. The thermal stability of this compound was analyzed by thermogravimetric analysis (TGA). As shown in Appendix A, the initial decomposition temperature of the compound was 247 °C, which may be caused by the organic components in this material. In addition, the second decomposition temperature of (TPA)_2_PbBr_4_ is 550 °C, which may be related to the evaporation of inorganic components in this compound [22].

Figure 2a shows the optical images of (TPA)_2_PbBr_4_ SCs, which is colorless in sunlight and shows blue emission under a 365 nm UV lamp. Subsequently, we explored the RT photophysical properties of (TPA)_2_PbBr_4_. As shown in Figure 2b, this compound has distinct bands at 325 nm and 385 nm in the PLE spectrum (monitored at 437 nm), and the excitation cutoff wavelength is 432 nm. Based on the electron transition rules in Pb^2+^ with 6s^2^ electron configuration, the two PLE bands can be attributed to the ^1^S_0_ → ^1^P_1_ and the ^1^S_0_ → ^3^P_n_ transitions, respectively [23]. Upon excitation at 350 nm, this compound shows a distinct emission band at 437 nm with a Stokes shift of 53 nm. Moreover, (TPA)_2_PbBr_4_ has a PLQE of 12%, and the corresponding CIE color coordinate (Figure 2c) is calculated to be 0.1555, 0.0544. The PL spectra under different excitation wavelength (300–400 nm) show a similar profile (Figure 2d), thus the observed blue emission in (TPA)_2_PbBr_4_ SCs stems from the intrinsic emission [24,25]. Combining previously reported Pb(II)-based metal halides with broad emission, the blue emission can be attributed to STEs [26,27,28]. Figure 2e shows that this compound has intense absorption bands at 274 nm and 353 nm at RT. In addition, the bandgap of (TPA)_2_PbBr_4_ was calculated to be 3.2 eV by Tauc equation. The RT decay lifetime of this compound was also measured by using a 365 nm picosecond pulsed laser, (Figure 2f), and the decay curve can be fitted by double exponential function of $I\left(t\right)=A_{1}exp\left(\;\tau /t_{1}\right)+A_{2}exp\left(\tau /t_{2}\right)$, where I is the luminescence intensity, τ represents the time after excitation, A is a constant and τ is the lifetime for the exponential component [29]. The decay curve contains a shorter-lived lifetime ($\tau_{1}$) of 1.8 ns and longer-lived lifetime ($\tau_{2}$) of 17 ns, which is comparable to other Pb(II)-based metal halides with STEs emission reported recently [30,31]. Moreover, the LT PLE spectrum has a distinct blue shift compared with RT PLE band out of the feeble electron-phonon coupling at low temperature, as shown in Appendix A. Subsequently, we compared the photophysical parameters of (TPA)_2_PbBr_4_ with other low-dimensional metal halides with blue emission (Appendix A), and we found that this compound has a small Stokes shift and low-energy light excitation. Therefore, we next study the causes of this phenomenon in detail.

We used DFT calculations to study the luminescence characteristics and electronic structure of (TPA)_2_PbBr_4_. Figure 3a shows the band structure of this compound. As can be observed, the valence band maximum (VBM) at Γ point and the conduction band minimum (CBM) at the Z point, indicating that (TPA)_2_PbBr_4_ is an indirect bandgap semiconductor. The calculated bandgap is 3.20 eV, which is slightly smaller than the experimental value of 3.58 eV. This may be due to the PBE bandgap error [32]. Moreover, the difference between the indirect gap and the direct gap is only ∼0.01 eV, illustrating the edge characteristics of the fluorescent band in the Brillouin region. Therefore, the electronic state of (TPA)_2_PbBr_4_ is highly localized. Figure 3b is a diagram of the total density of states (DOS) and orbital-resolved partial DOS, which shows that the highest occupied molecular orbital (HOMO) and lowest unoccupied molecular orbital (LUMO) are derived from the electronic state in [PbBr_4_]^2-^, but the former is composed of Br-4p and Pb-6s orbitals, and the latter has the mixed characteristics of Pb-6p and Br-4p. Figure 3c,d describe the electron distribution profiles of LUMO and HOMO in (TPA)_2_PbBr_4_, respectively. The electronic state of LUMO shows local characteristics, which is highlighted in [PbBr_4_]^2^^−^, and the electronic state of HOMO is mainly controlled by the Pb-p state. This is combined with the nearly flat band structure of VBM and CBM, which indicates that there is a strong quantum confinement effect in (TPA)_2_PbBr_4_. The distance between the nearest Pb-Pb in this compound is about 10.4 Å, so there is no electron band between the adjacent [PbBr_4_]^2−^ anions. Therefore, each [PbBr_4_]^2−^ can act as an independent luminescence center.

The structural deformation degree of the excited state is related to the change of Stokes shift [33,34], and then we calculated the structural deformation parameter of the ground state (GS) and excited state to explain the observed small Stokes shift in (TPA)_2_PbBr_4_. Figure 3e shows the change of bond length of [PbBr_4_]^2−^ in the GS and excited state, and the changes of bond angles and bond length of GS and excited state are listed in Appendix A, respectively. Particularly, most bond lengths in the calculated excited state of axial, such as Pb−Br2, Pb−Br3 and Pb-Br4, increase about 0.49−0.84% in comparison to the GS, while the bond length of Pb−Br1 is decreased by 1.09%. Compared with previously reported of (C_9_NH_20_)_2_SnBr_4_ [35], the degree of excited state structure distortion is greatly reduced. Meanwhile, the deformation parameter of (TPA)_2_PbBr_4_ was calculated by the Equation (1) [27]:(1)$$\Delta d=\left(\frac{1}{n}\right)\sum {\left[\frac{d_{n}-d_{ave}}{d_{ave}}\right]}^{2}$$
where *n* is the number of Pb−Br bonds, $d_{ave}$ is the Pb−Br average bond length, and $d_{n}$ is the length of each Pb−Br bond. According to the detailed bond length data in Appendix A, the deformation parameter (Δd) in the GS is 8.098 × 10^−4^, while the deformation parameter in the excited state is 5.299 × 10^−4^. The result shows that the excited state of (TPA)_2_PbBr_4_ is distorted, indicating that this compound has the potential to achieve broadband emission. However, its deformation parameters are one order of magnitude smaller than (TMA)_2_SbCl_5_·DMF [33], which may be the dominant reasons for (TPA)_2_PbBr_4_ having a small Stokes shift [36]. According to the detailed bond length data of Bmpip_2_PbBr_4_ and (C_13_H_19_N_4_)_2_PbBr_4_ SCs, the deformation parameter (Δ*d*) in the ground state is 24.3 × 10^−4^ and 19.7 × 10^−4^, respectively, which are larger than (TPA)_2_PbBr_4_ (8.098 × 10^−4^). Thus, the low distortion of [PbBr_4_]^2−^ species in (TPA)_2_PbBr_4_ may the dominant reason for the small Stokes shift.

In order to better understand the mechanism of the blue emission of (TPA)_2_PbBr_4_, the variable-temperature PL spectra (78–298 K) of this compound were measured excited by the 365 nm UV lamp, and the results are given in Figure 4a. Clearly, only one emission band can be observed in all temperature windows, which illustrates that (TPA)_2_PbBr_4_ has a single emission channel and stable phase structure. Then, the PL spectra of (TPA)_2_PbBr_4_ SCs at 78 and 298 K were fitted by Gaussian curves as shown in Appendix A. At 298 K, we can observe two distinct components, 427 nm (green line) and 452 nm (blue line), respectively. In addition, the emission bands at 427 nm and 452 nm can be attributed to the FEs and STEs, respectively [37]. However, PL spectrum at 78 K has a Gaussian shape, which should be assigned to FEs. The above conclusion can be demonstrated by temperature-dependent decay lifetime measurements of (TPA)_2_PbBr_4_ SC. The decay lifetime at 78 K (Appendix A) can be fitted via the mono-exponential function and the fitting decay lifetime is 6.68 ns, which can be ascribed to FEs emission, while the decay lifetime at 298 K contains a shorter-lived lifetime ($\tau_{1}$) of 1.8 ns and longer-lived lifetime ($\tau_{2}$) of 17 ns, which can be assigned to FEs and STEs emission, respectively. Moreover, the decay lifetime of FEs decreases from 6.68 ns to 1.8 ns with the increase in temperature, which can be attributed to the exciton fission to free carriers at high temperature, and the similar phenomenon was also found in CsPbX_3_ nanocrystals [38].

In Figure 4b, the position of the PL peak and the FWHM against temperature are plotted. With the temperature increase from 78 to 298 K, the peak position redshifts by 26 nm, which is similar to common semiconductors, such as CdS and ZnSe [39,40]. This phenomenon is possibly due to electron-phonon coupling causing the bandgap to shrink. Further, the FWHM is effectively widened with increasing temperature, similarly to other 0D organic metal halides, such as [BAPrEDA]PbCl_6_·(H_2_O)_2_ and (C_4_H_9_)_4_NCuCl_2_ [17,41]. In general, the enhanced coupling of electronic and acoustic phonons can be attributed to this phenomenon. In Figure 4c, we can observe the maximum PL intensity versus temperature. As the temperature rises from 78 to 198 K, the PL intensity of (TPA)2PbBr4 increases gradually. After the temperature rises from 198 to 298 K, the ratio begins to decrease. Traditionally, the strongest emission intensity can be observed at low temperature due to the prohibition of nonradiative recombination [42,43]. The unusual phenomenon observed in (TPA)_2_PbBr_4_ is due to the generation process of STE state and a large potential barrier (ΔE = KT = 17.1 meV). When the temperature is below 198 K, it is the production process of ste state, and its luminescence is affected by the phonon assisted tunneling effect. When the temperature reaches 198 K, its emission intensity is the strongest because the excited carriers larger than 17.1 mev cross the barrier and enter the self-trapped state. As the temperature increases beyond 198 K, the emission intensity weakens due to carrier scattering and thermal quenching. As for the similar process of FEs, the possible reason is that the lattice thermal expansion plays a key role at 78–198 K; thus, the FE intensity shows a linear increasing trend with the increase in temperature. When T > 198 K, the electron phonon interaction is enhanced, resulting in a large number of STE. In addition, the STE state dominates the luminescence, resulting in a decrease in the intensity.

The Huang–Rhys factor (S) closely correlates with the electron-phonon coupling strength and calculates its value using Equation (2) [44]:(2)$$\mathrm{FWHM}=2.36\sqrt{S}\hbar \omega_{phonon}\sqrt{\coth\frac{\hbar \omega_{phonon}}{2k_{B}T}}\;\;\;\;\;\;\;\;\;\;\;\;\;\;\;\;\;\;\;\;\;\;\;\;\;$$
where the *k*_*B*_ is Boltzmann constant, and *ω_phonon_* is phonon frequency. One can calculate that *S* is 7.098 by fitting the FWHM versus temperature (Figure 4d). The value is significantly higher than those reported for CdSe, ZnSe [45,46], and is comparable to other lead-free compounds [30,47]. Therefore, (TPA)_2_PbBr_4_ shows a high electron-phonon coupling. 

The Raman spectra of (TPA)2PbBr4 excited by 532 nm laser at 298 and 98 K are shown in Figure 4e. The Raman bands at 39, 51 and 101 cm^−1^ can be attributed the A_g_, B_1g_ and B_2g_ mode of the layered structure of PbBr_2_ [48]. Clearly, the Raman bands at high temperature move towards a low wavenumber, which may be caused by the lattice expansion [49], and 3D lead halide perovskites were found to exhibit the same phenomena [50]. Another interesting feature is that the Raman mode intensity decreases significantly at low temperature, which indicates that the electron–phonon interactions become stronger at high temperature. Notably, the modes at 101 and 309 cm^−1^ can be viewed as the overtone of B_1g_ mode. The Raman bands at 139 and 373 cm^−1^ are the sum-frequency of A_g_ and B_1g_ modes, because of 139 ≈ 39 + 2 × 51 and 373 ≈ 3 × 39 + 5 × 51. The Raman peak at 160 cm^−1^ should be the overtone of the B_1g_ and B_2g_ mode, with some contribution of acoustic phonon in this lattice, because 161 = 101 + 51 + 9. A peak near 79 cm^−1^ can be observed at low temperature. This is a two-phonon state, because it can reflect more phonon information at low temperature. Based on these spectral profile characters, we can conclude that this compound exhibits strong anharmonic electron-phonon interactions and further demonstrates the formation of STE in (TPA)_2_PbBr_4_.

Based on the above discussion, the observed blue emission in (TPA)_2_PbBr_4_ originates from the FEs and STEs, and its photophysical process can be inferred as shown in Figure 4f. Upon photoexcitation, [PbBr_4_]^2−^ clusters are excited from GS to excited state. FEs are produced first, and then they transit to self-trapped state. At low temperature, FEs can hardly overcome the energy barrier (ΔE) between the excited state and self-trapped state, so the observed emission is dominated by FEs. While at RT with the high thermal energy, the generated FEs can overcome the energy barrier and reach self-trapped state. At this time, we can observe the emission coexistence of FEs and STEs emission at RT. Overall, the external thermal energy plays an important role in regulating the photophysical properties of (TPA)_2_PbBr_4_.

## 4. Conclusions

In summary, a new monoclinic 0D organic lead bromide of (TPA)_2_PbBr_4_ SCs with *I2/a* symmetry was synthesized by the simple solution method. Interestingly, this compound shines a blue emission band at 437 nm with a relatively small Stokes shift of 53 nm and a FWHM of 50 nm. According to theoretical calculations based on DFT, we attributed this abnormally small Stokes shift to the small excited state structure distortion, that is FX coupled with the vibration introduced by amine. Moreover, the emission mechanism of (TPA)_2_PbBr_4_ were studied by temperature-dependent PL and Raman spectra, which reveals that the observed blue emission in (TPA)_2_PbBr_4_ stems from the FEs and STEs, and the population of FEs and STEs are affected by the external temperature. Our achievements provide some new ideas to design 0D organic−inorganic hybrid metal halides with blue emission under the near-UV-visible light excitation.

## Figures and Tables

**Figure 1 nanomaterials-12-02222-f001:**
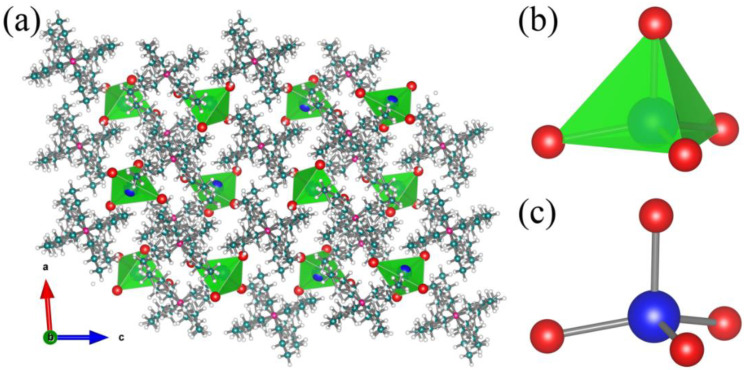
(**a**) Crystal structure of 0D (TPA)_2_PbBr_4_ (blue spheres: lead; red spheres: bromine; pink spheres: nitrogen; cyan spheres: carbon; white spheres: hydrogen). (**b**) View of individual tetrahedral clusters [PbBr_4_]^2−^. (**c**) Ball-and-stick diagram of an individual [PbBr_4_]^2−^ cluster.

**Figure 2 nanomaterials-12-02222-f002:**
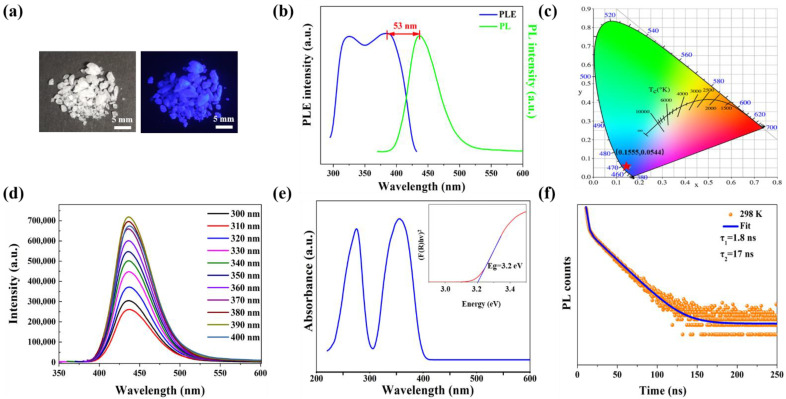
(**a**) Photographs of (TPA)_2_PbBr_4_ SCs under daylight and 365 nm illumination. (**b**) PL and PLE spectra of (TPA)_2_PbBr_4_ SCs measured at RT. (**c**) CIE color coordinates of (TPA)_2_PbBr_4_ SCs. (**d**) Excitation-dependent PL spectra of (TPA)_2_PbBr_4_ SCs. (**e**) Absorption spectrum, and the inset shows the Tauc plot of (TPA)_2_PbBr_4_ powder. (**f**) Decay lifetime of (TPA)_2_PbBr_4_ SCs measured at 298 K.

**Figure 3 nanomaterials-12-02222-f003:**
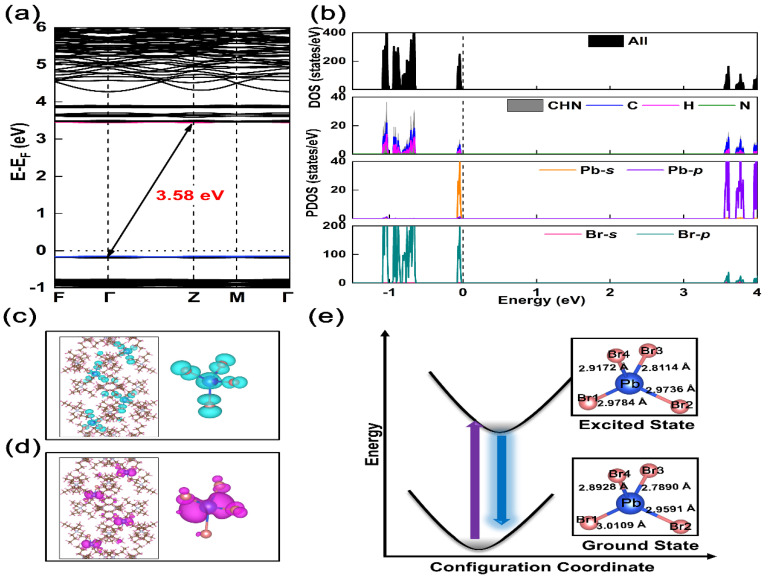
The electronic structure properties for (**a**) band structures and (**b**) DOS of (TPA)_2_PbBr_4_. The charge distribution density of LUMO (**c**) and HOMO (**d**) in (TPA)_2_PbBr_4_. (**e**) Schematic diagrams of the excited and ground states for [PbBr_4_]^2−^ structure with the specific bond lengths of Pb-Br, respectively.

**Figure 4 nanomaterials-12-02222-f004:**
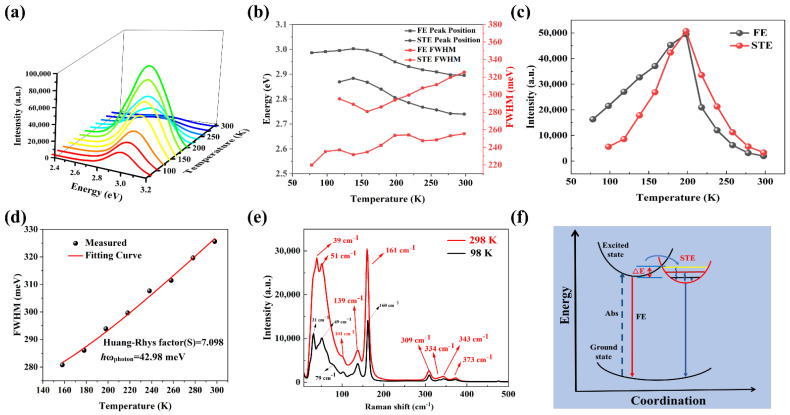
(**a**) Relationship between emission spectra and temperature of (TPA)_2_PbBr_4_. (**b**) Peak position; FWHM versus temperature. (**c**) PL intensity versus temperature. (**d**) FWHM obtained from PL spectra versus temperature. (**e**) The Raman spectra of (TPA)_2_PbBr_4_ SC at 298 K and 98 K, respectively. (**f**) Possible photophysical process of (TPA)_2_PbBr_4_.

## Data Availability

Not applicable.

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
