# Peer review of "A Zero-Dimensional Organic Lead Bromide of (TPA)2PbBr4 Single Crystal with Bright Blue Emission"

_nanomaterials, 2022, doi:10.3390/nano12132222_

Round 1

Reviewer 1 Report

The authors described the synthesis and the structural and optical characterization of a new 0D metal halide. They have performed DFT calculations to study the electronic structure of the material. The compound present a blue emission, with a relatively small Stokes shift, that the authors interpret as the emission of the combination of free exciton and self-trapped exciton. The results are interesting. However, I have several critics:

1 1   The authors stress the importance of 0D metal halide for optoelectronic applications in their introduction but the references cited (1-6) concern mainly 3D and 2D hybrid perovskites. The reference 15 concerns also a 2D perovskites and not a 0D metal halide, as the authors suggest. The introduction could be clearer on the current state of research specifically on 0D metal halide. The urgency to develop new blue emitting 0D metal halides is not obvious since the authors cite different previous studies on blue emitting 0D metal halide with high photoluminescence quantum yield. The authors explain that for material previously reported there is a problem of stability upon irradiation with UV.  The new compound they have synthetized can be excited at a 380 nm instead of 350 nm for the other materials. However, there is no evidence in the manuscript that this make a difference in term of photostability, all the more when compared with all inorganic 0D metal halide.   

   2 In this study, the 0D metal halide material is based on a tetrahedral structure, which, to my knowledge is relatively rare. Most of the compounds in the literature are based on an octahedral structure. However, this fact seems insufficiently highlighted in the manuscript. The results, regarding the impact of the distortion for example, are compared indistinctly with compounds presenting an octahedral structure or a tetrahedral structure.  A previous report on a 0D lead bromide with similar PbBr42- cluster, with a blue emission and small Stokes shift, is cited hastily (ref 17) but the results and interpretation of the authors are not compared to this previous study, despite the similarity of the material and observations.

3   3 The authors have measured the photoluminescence excitation spectrum of the material. Two peaks are visible. The authors claim that they can be assigned to 1P1 à1S0 and 3Pn à1S0 transitions based on transition rules for Pb2+. The reasoning is unclear. The author also exctract the bandgap from the absorption spectrum. Later, the authors calculate the band structure of the compound. They explain then that the electronic states are based on Pb and Br orbitals.

So, it seems that the optical transition observed in PLE and absorption should correspond to transition between the valence band and conduction band?

Author Response

We are grateful for the important comments of reviewers, we have modified our manuscript in details according to reviewers’ comments. Main modifications were emphasized through using red color fonts in the revised manuscript. For the Reviewers’ comments, we have given specific answers.

Reviewer 2 Report

The authors proposes an interesting papers on 0D monoclinic 0D organic lead bromide of (TPA)2PbBr4 SCs as blue emitter under near UV excitation al small Stokes shift. They describe the luminescence of the(TPA)2PbBr4 SC as of excitonic nature (FE and STE)of [PbBr4]2 in a distorted environment . Good comparison of the material of the SC with similar materials in the literature are presented to support the better performance of the proposed  phosphor.

Here I propose the following clarification before publication.

1) in general in TRPL and PL/PLE spectra clarify the excitation WL and the emission WL (Fig 2, Fig S3....)

2) The numerical analysis of the PL spectra in gaussian components should be performed according to the following paper J. Phys. Chem. Lett. 2013, 4, 19, 3316–3318

3) Line 211-212, clarify the sentence also considering the description in line 143-144

4) Fig 2b has STE and FE bands the same PL features? can the authors perform PLE at LT?

5) L 213 I am not getting the point. Indeed how have the authors measured the TRPL? How can the authors discriminate the STE and the FE component at RT during TRPL? In other words, where have the authors collected the emissions for the TRPL at RT? They should have also considered the overlapping of the 2 bands...

6) STE band lifetime is then non affected by the temperature (T)? However it disappeared at LT...or it seems...

7) following the point 6), how have the authors monitored the peak intensity and FWHM and position? I strongly  suggest to perform the analyses according to  point 2) (gaussian components in eV!!) and once established the Xcenter in eV + FWHM eV both at the RT  and LT, please give the FWHM, POSTION and INTENSITY of the 2 bands following the behaviours of the 2 bands  as a function of the T.  Only in this way one can distinguish the FE behaviour from the STE band (that is the main topic of the work)

Author Response

(The authors gave the same response as above.)

Round 2

Reviewer 1 Report

The authors have provided detailed answers to my comments. I recommend the manuscript for publication in Nanomaterials.

Author Response

We are grateful for the important comments of reviewers, we have modified our manuscript in details according to reviewers’ comments. For the Reviewers’ comments, we have given specific answers.

Reviewer 2 Report

The authors have replied to my comments in details, thanks

However figure S5 ESI

I cannot distinguish the experimental data from the Fit

Please make the graph clear with exp. data + gaussian components+ cumulative fit

Author Response

(The authors gave the same response as above.)
